# Broad-Spectrum *Salmonella* Phages PSE-D1 and PST-H1 Controls *Salmonella* in Foods

**DOI:** 10.3390/v14122647

**Published:** 2022-11-27

**Authors:** Yajie Cao, Runwen Ma, Ziyong Li, Xinyu Mao, Yinan Li, Yuxin Wu, Leping Wang, Kaiou Han, Lei Li, Dongxin Ma, Yuqing Zhou, Xun Li, Xiaoye Wang

**Affiliations:** 1College of Animal Science and Technology, Guangxi University, Nanning 530004, China; 2Guangxi Zhuang Autonomous Region Engineering Research Center of Veterinary Biologics, Nanning 530004, China; 3Guangxi Key Laboratory of Animal Reproduction, Breeding and Disease Control, Nanning 530004, China; 4Guangxi Colleges and Universities Key Laboratory of Prevention and Control for Animal Disease, Nanning 530004, China

**Keywords:** *Salmonella*, broad-spectrum bacteriophages, food contamination, biocontrol

## Abstract

Food contamination by *Salmonella* can lead to serious foodborne diseases that constantly threaten public health. Innovative and effective strategies are needed to control foodborne pathogenic contamination since the incidence of foodborne diseases has increased gradually. In the present study, two broad-spectrum phages named *Salmonella* phage PSE-D1 and *Salmonella* phage PST-H1 were isolated from sewage in China. Phages PSE-D1 and PST-H1 were obtained by enrichment with *Salmonella enterica* subsp. *enterica* serovar Enteritidis (*S.* Enteritidis) CVCC1806 and *Salmonella enterica* subsp. *enterica* serovar Typhimurium (*S.* Typhimurium) CVCC3384, respectively. They were able to lyse *Salmonella*, *E. coli* and *K. pneumoniae* and exhibited broad host range. Further study demonstrated that PSE-D1 and PST-H1 showed high pH and thermal tolerances. Phage PSE-D1 belongs to the *Jiaodavirus* genus, *Tevenvirinae* subfamily, while phage PST-H1 belongs to the *Jerseyvirus* genus, *Guernseyvirinae* subfamily according to morphology and phylogeny. The results of genome analysis showed that PSE-D1 and PST-H1 lack virulence and drug-resistance genes. The effects of PSE-D1 and PST-H1 on controlling *S.* Enteritidis CVCC1806 and *S.* Typhimurium CVCC3384 contamination in three kinds of foods (eggshells, sausages and milk) were further investigated, respectively. Our results showed that, compared to phage-free groups, PSE-D1 and PST-H1 inhibited the growth of their host strain significantly. A significant reduction of host bacteria titers (1.5 and 1.9 log_10_ CFU/sample, *p* < 0.001) on eggshells was observed under PSE-D1 and PST-H1 treatments, respectively. Furthermore, administration of PSE-D1 and PST-H1 decreased the counts of bacteria by 1.1 and 1.2 log_10_ CFU/cm^2^ (*p* < 0.001) in sausages as well as 1.5 and 1.8 log_10_ CFU/mL (*p* < 0.001) in milk, respectively. Interesting, the bacteriostasis efficacy of both phages exhibited more significantly at 4 °C than that at 28 °C in eggshells and milk and sausages. In sum, the purpose of our research was evaluating the counteracting effect of phage PSE-D1 and PST-H1 on the spread of *Salmonella* on contaminated foods products. Our results suggested that these two phage-based biocontrol treatments are promising strategies for controlling pathogenic *Salmonella* contaminated food.

## 1. Introduction

The primary cause of foodborne illnesses is bacterial contamination in food. *Salmonella* accounts for the highest proportion of bacterial food-borne contamination in the U.S.A, causing 1.4 million cases of *Salmonella* food-borne diseases each year [1]. During 2017, it was reported in Britain that *S.* Typhimurium and *S.* Enteritidis were two of the top five most common serotypes in bacterial infections [2]. Fresh-cut produce that is easily consumed, uncooked and under-cooked meat, poultry, and eggs contaminated with *Salmonella* are further risk factors for infection [1]. *Salmonella* contamination was very common in eggs, meat, and meat products [3].

Effective and innovative methods to preserve food are critical for food quality and safety. Bacterial food contamination can be controlled by chemical, physical, and biological techniques. Nevertheless, some traditional procedures cause adverse effects on food quality [4]. Thermal treatment, the most widely used procedure for microbial inactivation in foods, usually causes unwanted side effects in the sensory and nutritional qualities of food [5]. In the germicidal process, irradiation can change the flavor and color of meat products in addition to producing a distinctive odor. Although antimicrobials like potassium benzoate and sodium lactate are frequently used for increasing shelf-life, the safety of foods cannot be guaranteed if they are used alone [6]. In order to reduce the occurrence of food-borne salmonellosis and inhibit the contamination of *Salmonella* in foods, appropriate approaches are urgently required.

Phage-based biological control strategies are considered potential methods to control bacterial contamination of food gradually. Phages are the most abundant microorganisms found on earth and are widespread on foods of various origins [7]. Compared to conventional antimicrobials, bacteriophages are widespread, easily available, and harmless to the public. Phages have high specificity and an effective ability to lyse targeted pathogenic bacteria. They are natural killers of bacteria. Until now, some studies have reported the application of bacteriophages on bacterial contamination in dairy foods. Huang et al. showed that broad-spectrum *Salmonella bacterio* phage LPST10 had the potential to reduce titers of *S.* Typhimurium and *S.* Enteritidis by 0.92 to 5.12 log_10_ CFU/sample in lettuce and sausages [8]. Another similar study certified that, in both liquid egg white and yolk, considerable bacteria inhibition in two MDR *Salmonella* strains were observed by phage D1-2 at different temperatures [9]. Furthermore, Hudson et al. found that, compared to phage-free groups, phages could significantly inhibit the growth of E. coli O157:H7 by 4 log_10_ CFU/cm^2^ on cooked and raw beef at two different temperatures [10]. These studies suggested that phages have the potential to control bacterial food-borne contamination.

Therefore, our paper aimed to investigate the effectiveness of bacteriophages on inhibiting *S.* Enteritidis CVCC1806 and *S.* Typhimurium CVCC3384 contamination in eggshells, chicken sausages, and milk at different temperatures to explore their use in these and other food products.

## 2. Materials and Methods

### 2.1. Bacteria Strains and Culture Medium

Our study involved seventy bacteria strains, including 6 *Salmonella* strains, 48 *E*. *coli* strains, 15 *K*. *pneumoniae* strains, and 1 *P. aeruginosa* strain. *Salmonella enterica* subsp. *enterica* serovar Enteritidis (*S.* Enteritidis) CVCC1806, *Salmonella enterica* subsp. *enterica* serovar Typhimurium (*S.* Typhimurium) CVCC3384, *Salmonella enterica* subsp. *enterica* serovar Pullorum (*S.* Pullorum) SX-1014, and SF-0923 were bought from the China Veterinary Culture Collection Center (CVCC). The remaining strains came from poultry and pig farms or clinics. In our lab, all bacteria strains were stored at −80 °C in LB broth contained with 20% *v*/*v* glycerol.

### 2.2. Isolation and Purification of Bacteriophages

Two phages named *Salmonella* phage PSE-D1 and *Salmonella* phage PST-H1 were isolated with *S.* Enteritidis CVCC1806 and *S.* Typhimurium CVCC3384 as host bacteria, respectively. PSE-D1 and PST-H1 were separated from cattle farm sewage in Guangxi, China. Purification and amplification of the two phages were then performed by using host strains, and 10mL of LB broth was added into the mixture of 100 μL samples and 100 μL log-phase bacteria and subsequently cultured for 4–5 h at 37 °C. The supernatant was collected by filtering through a micro-porous membrane of 0.22 µm after centrifuging at 12,000 rpm for 5 min. The morphology of phage plaque and titer were investigated by using the double-layer agar method. The presence of plaque would be observed after incubating for 4–5 h at 37 °C in an incubator [11,12]. The purified phage suspension was obtained after purification three times. Finally, phages would be preserved at −80 °C into a SM buffer containing 20% *v*/*v* glycerol in our laboratory. 

### 2.3. Host Range

The host spectrum of phage PSE-D1 and PST-H1 were investigated against 6 *Salmonella* strains, 48 *E*. *coli* strains, 15 *K*. *pneumoniae* strains, and 1 *P*. *aeruginosa* strain through a spot test [13]. The bottom layer consisted of LB broth and 1.0% agar (10 mL in total). The mixture of LB contained 0.5% agar (2–3 mL), and the bacterial culture of the tested bacterial strains (100 µL) was overburden. The overlay was added after solidification of the bottom layer. Phage suspensions (4–5 µL, 10^9^ PFU/mL) were then dropped onto the previously mentioned plates and incubated at 37 °C overnight. 

### 2.4. Characterization of Phages

#### 2.4.1. One Step Growth

A one-step growth curve experiment of phages was determined with some modifications [14]. A mixture of log-phase bacteria and a phage was incubated for 15 min at 37 °C. The supernatant fluid was then taken out after centrifuging at 12,000 rpm for 1 min and resuspended with LB broth. Then the suspension was transferred into 10 mL LB broth and incubated at 37 °C. Sampling was done at 0, 10, 20, 30, 40, 50, 60, 70, 80, 90, 100, 110, and 120 min and used the double-layer method to assess phage titers. Burst size was calculated according to standard methodology [15].

#### 2.4.2. Lytic Capacity of Phages under Different Multiplicity of Infection (MOI)

Culture *S.* Enteritidis CVCC1806 and *S.* Typhimurium CVCC3384 strains were incubated under the above conditions [16]. The culture was then loaded into glass test tubes at 1 × 10^6^ CFU/mL, and phages were added under MOIs of 100, 10, 1, 0.1, or 0.01 (PFU/CFU). Incubated for 15h at 37 °C in a table concentrator at 180 rpm and the cellular density was evaluated at OD_450nm_ every hour. 

#### 2.4.3. Thermal and pH Stability of Phages

The thermal and pH stability of the phages were determined using a modification of the previously described methods [17,18,19]. For temperature sensitivity testing, a purified phage was incubated at refrigeration temperature (4 °C), room temperature (25 °C), incubation temperature (37 °C), and other temperatures (50, 60, 70, 80 °C), respectively. Sampling was performed at 30 and 60 min, and phage titers were assessed by double-layer method. The purified phages were incubated at different pH values (2–13) for 2 h at 37 °C in order to evaluate the pH sensitivity of the phages. 

#### 2.4.4. Electron Microscopy

After being resuspended in ammonium acetate (0.1 mol/L), the phage suspension (10^10^ PFU/mL) was centrifuged at 4 °C with 40,000 rpm for 1 h. The immersed phages were suspended in the copper grid for 10 min, followed by staining with a 2% volume fraction of phosphotungstic acid solution (pH = 7) for 10 min. The morphology of the phages was observed by transmission electron microscopy (TEM) (Hitachi High-Tech Co., Ltd., Tokyo, Japan). 

#### 2.4.5. Extraction and Analysis of Phage Genome Sequence

A suspension with a high titer (1010 PFU/mL) of phages was used to extract genomic DNA with modification of the previously described methods [20]. Beijing University of Chemical Technology carried out the control of sequencing data quality and the construction of a sequencing library. A RAST server 2.0 was used to forecast and annotate the coding DNA sequences (CDSs). The Virulence Factor Database and Comprehensive Antibiotic Resistance Database were used to screen the hypothetical virulence factors and drug-resistance genes, respectively [21,22]. A CGView Server was used to depict a comparative circular genome map of the phage genome. The phylogenetic trees were constructed by using the ClustalW program based on the whole genome in MEGA-X [23].

### 2.5. Assays on the Surface of Egg Shells, Chicken Sausage and in Fresh Milk

#### 2.5.1. Sample Preparation

*Salmonella* CVCC1806 and CVCC3384 inoculum were prepared as mentioned above. Eggs, chicken sausages, and fresh milk were bought from a supermarket and stored at 4 °C. To eliminate bacteria on the surface of the eggs, the surface was washed with 75% ethanol and exposed to ultraviolet radiation for 40 min. Packed chicken sausages were aseptically cut into slices with a diameter of 2cm and a thickness of 1cm in a biological safety cabinet. To reduce the indigenous bacteria level before inoculation, the pieces of chicken sausages were irradiated with UV light for 1h in a safety cabinet (30 min on each side). 

#### 2.5.2. *Salmonella* and Phage Preparation 

The host strain cultures were inoculated and spread evenly on the surface of the eggshells (100 µL, 1.0 × 10^8^ CFU/mL) and sausages (25 µL, 4.0 × 10^5^ CFU/mL). Before phage treatment, all samples were dried for 10 min (eggs) or 15–20 min (sausages) in a biological safety cabinet. The *Salmonella* suspension (100 µL, 1.0 × 10^5^ CFU/mL) was artificially inoculated into the fresh milk. 

Samples were inoculated with a phage lysate at a MOI of 100 and 10, respectively at 4 °C and 28 °C for phage treatment. The above phage lysate was suspended in Tris-SM buffer, and sterile Tris-SM buffers were added into the control samples. The microbiological analysis of eggshells and sausages were carried out at 2, 4, 6, 24, and 48h [24,25], and the milk was carried out at 2, 4, 6, and 24h [26,27]. The eggshells were washed with physiological saline [28], and sausages were ground before counting. Plating on *Salmonella* Shigella agar (SS agar) was used to calculate bacterial counts (CFU/mL).

### 2.6. Statistical Analysis 

All the experiments were conducted three times. SPSS statistical software was used for analysis statistical. The significance level (*p* < 0.05) between each group was determined by a one-way ANOVA with a confidence interval of 95%. 

## 3. Results

### 3.1. Isolation and Purification of Bacteriophages

Two broad-spectrum phages named *Salmonella* phage PSE-D1 and *Salmonella* phage PST-H1 were isolated from sewage. Phage PSE-D1 and PST-H1 were obtained by enrichment with *S.* Enteritidis CVCC1806 and *S.* Typhimurium CVCC3384, respectively. The plaque of these bacteriophages on the double-layer agar plate was bright and clean, with clear edges. (Figure 1). 

### 3.2. Host Range 

A total of 70 strains were used to test the hosts spectrum of phage PSE-D1 and phage PST-H1, respectively, including 6 *Salmonella* strains, 48 *E*. *coli* strains, 15 *K*. *pneumoniae* strains, and 1 *P*. *aeruginosa* strain (Table 1). Phage PSE-D1 could infect 3 *Salmonella* strains, 13 *E. coli* strains, and 8 *K. pneumoniae* strains. Phage PST-H1 could infect 3 *Salmonella* strains, 10 *E. coli* strains, and 11 *K. pneumoniae* strains.

### 3.3. Biological Characterization of Phages

#### 3.3.1. Electron Microscopy

According to the International Committee on Taxonomy of Viruses (ICTV) and structural analysis by transmission electron microscopy, PSE-D1 belongs to the *Tevenvirinae* subfamily. The head of PSE-D1 is elongated, icosahedral-shaped, and the diameter is about 69.2 nm. The tail of PSE-D1 is contractile with a 94.5 nm length approximately (Figure 2a). Phage PST-H1 belongs to the *Guernseyvirinae* subfamily. PST-H1 has a typical regular icosahedral-shaped head, and the diameter is about 51.0 nm. The tail of PST-H1 is long and non-contractile with a 125.1 nm length approximately (Figure 2b).

#### 3.3.2. One Step Growth

The infection potential of the phages was analyzed through one-step growth curves (Figure 3). PSE-D1 had a latent period of 10 min approximately, according to the growth curve (Figure 3a). Over the next 60 min, the phage titer of PSE-D1 significantly increased, and PSE-D1 then grew stably until 120 min. The burst size of PSE-D1 was around 110 PFU/CFU. PST-H1 had a latent period of 20 min approximately (Figure 3b) and had a burst size of 183 PFU/CFU. 

#### 3.3.3. Lytic Capacity of Phages against Host at Different MOI

The growth of the host (OD450 nm) was measured to determine the lytic capacity of the phage against strain. As shown in Figure 4, proliferation of the host strains could consistently be inhibited by phages PSE-D1 and PST-H1 for 15 hours, respectively. The viable counts of host strains increased at 1–15 h in the positive control groups. The efficacy at the MIOs of 100 and 10 were better than other treatment groups with lower-titer phages, revealing that the antibacterial effect of PSE-D1 was MOI dependent. PST-H1 had an obvious strong inhibitory effect of CVCC3384 within 5 h. The strain counts of phage-treatment groups reduced significantly at 1-15 h compared to the phage-free control (Figure 4b). 

#### 3.3.4. Thermal and pH Stability of Phages

Phages PSE-D1 and PST-H1 were stable at 4–60 °C in a thermal stability test, revealing high thermal tolerance, and phage titers were undetectable after 60 min at 80 °C. At 70 °C, phage titers kept decreasing (Figure 5a,b). For the pH stability test, PSE-D1 and PST-H1 showed obvious stability at pH 3-11 (Figure 5c,d). At pH 2, pH 12, or pH 13, PSE-D1 and PST-H1 remained inactivated.

### 3.4. Genomic Analysis of Phages

The Gen Bank accession of PSE-D1 was no. ON550260, and that of PST-H1 was no. ON454038. The two phages were both double-stranded DNA. PSE-D1 had 166,604 bp and the GC content was 39.59% (Figure 6a). PST-H1 consisted of 42,036 bp, and the GC content was 49.95% (Figure 6b). While 115 CDSs had been predicted to encode functional proteins, there were 266 CDSs in the PSE-D1 genome totally. A total of 62 CDSs were associated with DNA replication and regulation; 48 CDSs participated in packing and morphogenesis proteins; 5 CDSs encoded proteins of host lysis among 115 CDSs (Appendix A). A total of 61 CDSs were founded in PST-H1 genomes according to protein coding genes prediction, of which 16 CDSs were predicted to encode functional proteins. Among 16 CDSs, 6 CDSs encoded the replication and regulation of DNA, 1 CDS encoded lysis protein, and 9 CDSs encoded packing and morphogenesis proteins (Appendix A). 

The plaques of phage PSE-D1and PST-H1 had typical characteristics of lytic bacteriophages. Furthermore, gene clusters like Cro, CI, C2, C3, N, and Q were absent from the phage genome, which related to lysogeny, indicating that PSE-D1and PST-H1 were lytic bacteriophages [29]. No known genes associated with virulence and antibiotic-resistance were found in the genome sequence of phages, indicating that PSE-D1 and PST-H1 are safe for use in phage therapy. 

PSE-D1 encoded 15 CDSs associated with tail fiber proteins, and PST-H1encoded 3 CDSs related to tail fiber proteins, which related to bind with bacterial receptors. SDS 162 of PSE-D1 encoded straight tail fiber (short tail fiber), mediating the adsorption of the phage and host strain. However, SDS162 had no specificity for bacteria adsorption [30]. Long tail fibers contributed to the host cell specificity, while CDS 244 of PSE-D1 encoded a long tail fiber proximal subunit, which related to adhesion between bacteria and bacteriophages [30]. Complete genome sequence analysis revealed that phage PSE-D1 was highly homologous with Klebsiella phage Vb-KpnM-FRZ284 (98% coverage), *Klebsiella* phage JD18 (98% coverage), and *Klebsiella* phage KP1 (97% coverage). PST-H1 was highly homologous with the following phages: *Salmonella* phage S55 (93% coverage), *Salmonella* phage LP31 (93% coverage), and *Salmonella* phage CKT1 (91% coverage) according to BLAST analysis. 

Based on the whole genome, phylogenetic trees were created to further analyze the relationship between PSE-D1 and PST-H1 and other phages belonging to the *Caudovirales* order, respectively (Figure 7). The results of phylogenetic relationships revealed that the *Klebsiella* phage KP1 which belongs to the *Jiaodavirus* genus and *Tevenvirinae* subfamily, was the closest to phage PSE-D1. Phage PST-H1 shared a close relationship to *Salmonella* phage S55, which belongs to the *Jerseyvirus* genus, *Guernseyvirinae* subfamily.

### 3.5. Application of Phages in Different Foods

#### 3.5.1. *Salmonella* Reduction on Egg Shells

At both 4 °C and 28 °C, the viable counts of *S.* Enteritidis CVCC1806 with PSE-D1 treatment reduced significantly (*p* < 0.05, Figure 8a,b) after 48 h in eggshells. At 4 °C, compared with the positive control, PSE-D1 significantly inhibited the growth of CVCC1806 on eggshells at a MOI of 100 (1.5 log_10_ CFU/sample, *p* < 0.001) and a MOI of 10 (1.1 log_10_ CFU/sample, *p* < 0.05) at 48h (Figure 8a). Compared to the positive control, PSE-D1 showed a significant inhibition effect on the growth of CVCC1806 on eggshells at a MOI of 100 (1.0 log_10_ CFU/sample) and a MOI of 10 (0.9 log_10_ CFU/sample) at 48 h at 28 °C (*p* < 0.05, Figure 8b). At 4 °C and 28 °C, the group with PSE-D1 at a MOI of 100 showed better bactericidal efficiency than the group at a MOI of 10. At 4 °C, PST-H1 achieved a 1.9 log_10_ CFU/sample (*p* < 0.001) reduction of *S.* Typhimurium CVCC3384 at a MOI of 100 and 1.4 log_10_ CFU/sample (*p* < 0.001) reduction when at a MOI of 10 on 48 h, respectively (Figure 8c). At 28 °C, the CVCC3384 was reduced by more than 1.4 log_10_ CFU/sample at 48h at both a MOI of 100 and 10 (*p* < 0.001, Figure 8d), indicating a strong bactericidal effect of PST-H1 against CVCC3384 on eggshells. No *Salmonella* was detected from the negative control samples. 

#### 3.5.2. *Salmonella* Reduction on Sausages

At 4 °C, compared to the phage-free group, PSE-D1 significantly inhibited the growth of CVCC1806 on sausages at a MOI of 100 (1.1 log_10_ CFU/cm^2^, *p* < 0.001) and a MOI of 10 (1.1 log_10_ CFU/cm^2^, *p* < 0.001) at 48h (Figure 9a). At 28 °C, PSE-D1 achieved 0.6 log_10_ CFU/cm^2^ (*p* < 0.001, Figure 9b) reduction on CVCC1806 with a MOI of 100 at 48 h. At 4 °C, the viable counts of CVCC3384 decreased (0.9 log_10_ CFU/cm^2^, 1.0 log_10_ CFU/cm^2^) with PST-H1 treatment at a MOI of 100 and 10, respectively (*p* < 0.001, Figure 9c). PST-H1 at a MOI of 100 and 10 achieved reduction (0.8 log_10_ CFU/cm^2^, 0.7 log_10_ CFU/cm^2^, *p* < 0.001) in CVCC3384 at 48 h at 28 °C, respectively (Figure 9b). Within 48 h, the viable counts of CVCC3384 decreased (1.2 log_10_ CFU/cm^2^, 0.9 log_10_ CFU/cm^2^, *p* < 0.001) with PST-H1 treatment at 4 °C and at 28 °C, respectively (Figure 9c,d). No *Salmonella* was detected from the negative control samples. 

#### 3.5.3. *Salmonella* Reduction in Milk

At 4 °C, compared to the positive control, PSE-D1 had a remarkable inhibitory effect on CVCC1806 growth (1.5 log_10_ CFU/mL, 1.2 log_10_ CFU/mL, *p* < 0.001) when at a MOI of 100 and 10 in 24 h in milk, respectively (Figure 10a). At 28 °C, the viable counts of CVCC1806 decreased to a low level with PSE-D1 at a MOI of 100 and 10 (reduction of 1.0 log_10_ CFU/mL, 0.7 log_10_ CFU/mL, *p* < 0.001, Figure 10b). After 24 h of incubation at both 4 °C and 28 °C, treatment with PST-H1 showed remarkable reductions of CVCC3384 counts in contaminated milk (p < 0.001, Figure 10c,d). PST-H1 obviously inhibited CVCC3384 at 4 °C and 28 °C (reduction of 1.8 log10 CFU/mL, 1.6 log_10_ CFU/mL, *p* < 0.001) after 24 h incubation, respectively (Figure 10c,d). No *Salmonella* was detected from the negative control samples. 

## 4. Discussion

In this study, two phages named *Salmonella* phage PSE-D1 and *Salmonella* phage PST-H1 were isolated from cattle farm sewage in Guangxi, China. Phage PSE-D1 could infect 3 of 6 *Salmonella* strains, 13 of 48 *E. coli* strains, and 8 of 15 *K. pneumoniae* strains tested in this paper. A total of 3 *Salmonella* strains, 10 *E. coli* strains, and 11 *K. pneumoniae* strains were all lysed by phage PST-H1. *Salmonella* and *E. coli* are common bacterial foodborne pathogens, which are often found in contaminated dairy products, eggs, chicken, and chicken products. *S.* Typhimurium and *S.* Enteritidis collections were chosen to evaluate the host spectrum because they are the most commonly reported serovars that affect the food and veterinary industry (specifically poultry), suggesting that the two phages have the potential to inhibit *Salmonella* from contaminating food.

One-step growth curves reveal the infection potential of each phage. The latent period of PSE-D1 was around 10 min, and PST-H1 had a latent period of 20 min approximately. Their latent periods were shorter than that of many other *Salmonella* phages that have been reported [31,32,33]. The previous study revealed that shorter latent periods can substantially reduce phage generation times, making them more suitable for biocontrol applications. The burst size of PSE-D1 is 110 PFU/CFU, while that of PST-H1 is 183 PFU/CFU, which is higher than many other reported phages [8,31,34].

PSE-D1 and PST-H1 are highly stable at 60 °C for 1h, implying that they could be utilized to pasteurize dairy products. In line with the previous studies, PSE-D1 and PST-H1 exhibited high pH tolerance by remaining active under pH ranging from 3 to 11. The two phages can be employed in food with varying pH, since they are stable under a wide range of acidic and alkaline circumstances. The high stability of phage PSE-D1 and PST-H1 at temperature and pH might allow PSE-D1 and PST-H1 to be stored in food for a longer time.

For simulating the storage conditions of food products at different temperatures, we chose 4 °C and 28 °C to represent cold storage and room temperature, respectively. *Salmonella* counts with the phage treatment decreased significantly (*p* < 0.05) on the surface of egg shells (1.9 log_10_ CFU/sample), sausages (1.2 log_10_ CFU/cm^2^), and in milk (1.8 log_10_ CFU/mL) compared to phage-free groups after 48h at two different temperatures (4 °C and 28 °C). Temperature largely influenced the effectiveness of phage PSE-D1 and PST-H1. It is reported that phages had greater bactericidal capacity when refrigerated [8,9,35]. A similar result was observed in this study. At 4 °C, PSE-D1 and PST-H1 showed a stronger inhibitory effect on *Salmonella* growth than that at 28 °C. PSE-D1 and PST-H1 can maintain high activity for a long time at 4 °C. The phage titer in Chinese cabbage was dramatically lower after incubation at 25 °C for 24 hours than that at 4 °C in the previous study, demonstrating that storage at room temperature may also impair the stability of the phage [36]. *Salmonella* activity was relatively lower at 4 °C than that at 28 °C, indicating that the phages had a better bacteriostasis effect at 4 °C.

The quality and safety of milk are highly valued in the dairy industry. Milk might be an important medium for *Salmonella* transmission to humans [37]. In raw milk, *Salmonella* is one of the most common bacterial foodborne pathogens especially in developing countries [35]. The existence of *Salmonella* is the lead cause of contamination after pasteurization in dairy products [38]. Therefore, controlling the contamination after milk processing is a key factor to prevent the transmission of *Salmonella* into dairy products. Because *Salmonella* is widely spread in chickens, it is important to reduce the incidence of *Salmonella* contamination in chicken and chicken products during their production, transportation and processing [39]. Consumption of shell eggs has been associated with *S.* Enteritidis infections in humans in the U.S.A. With the reduction of pathogens on eggshells, the possibility of cross-shell contamination becomes less [40]. The application of salmonella bacteriophage products in meat and poultry products has been authorized [41]. There are few reports on the direct application of phages on eggshells, while phages PSE-D1 and PST-H1 showed 1.5 log_10_ CFU/sample reduction of *Salmonella* on the surface of eggshells with significant difference (*p* < 0.001) occurring in 48h. It is reported that phages had good applications in chicken and milk [37,38,39]. In this study, phages PSE-D1 and PST-H1 showed good bactericidal effect in chicken sausages and milk with significant difference (*p* < 0.001), indicating that the two phages had potentiality for being used in controlling *Salmonella* contaminated food.

The bacteriostatic effects of PSE-D1 and PST-H1 were found to be more effective with phages with higher titers. At 4 °C, PSE-D1 treatment with different counts (10^10^ PFU/sample, 10^9^ PFU/sample) lead to a significant decrease of bacteria counts (1.5 log_10_ CFU/sample, 1.1 log_10_ CFU/sample, *p* < 0.05) in CVCC1806 contaminated eggshells. At 4 °C, the viable counts of CVCC3384 decreased (1.9 log_10_ CFU/mL, 1.4 log_10_ CFU/mL, *p* < 0.001) with phages treatment with different titers (10^10^ PFU/mL, 10^9^ PFU/mL) in eggshells. The previous study revealed that direct contact between bacteria and bacteriophages is more likely to occur at higher concentrations [14,42]. High concentrations of phage increased the probability of encountering *Salmonella* cells [24,25]. 

## 5. Conclusions

Our study described two broad-spectrum phages, PSE-D1 and PST-H1, isolated from cattle farm sewage, exhibiting a strong lytic effect on *S.* Enteritidis CVCC1806 and *S.* Typhimurium CVCC3384. They showed high pH and thermal tolerances as well as short latent periods. Additionally, PSE-D1 and PST-H1 showed considerable inhibition of growth of *S.* Typhimurium and *S.* Enteritidis in LB broth and three different foods, respectively. No known virulence factors or antibiotic-resistance related genes were found in PSE-D1 and PST-H1 genomes, indicating they are a promising bactericide during food preservation.

## Figures and Tables

**Figure 1 viruses-14-02647-f001:**
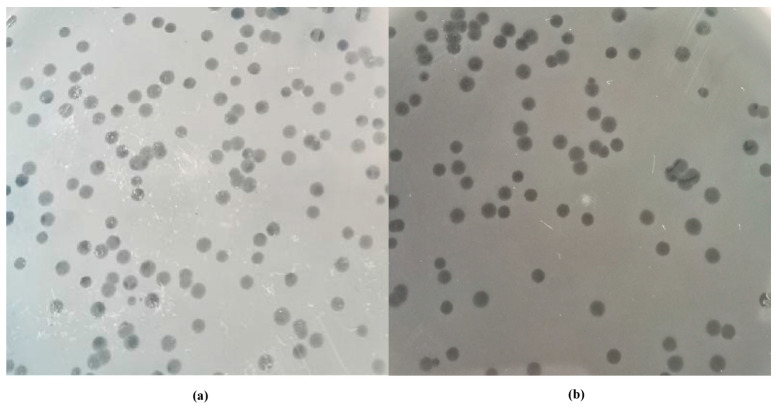
Morphology on double−layer agar plates. (**a**) phage PSE-D1. (**b**) phage PST-H1.

**Figure 2 viruses-14-02647-f002:**
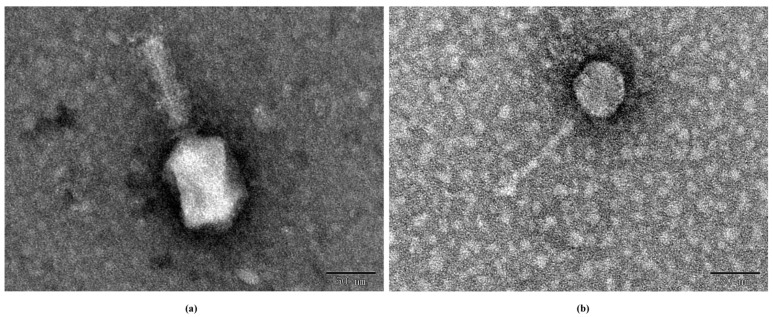
TEM of phages. (**a**) phage PSE-D1. (**b**) phage PST-H1.

**Figure 3 viruses-14-02647-f003:**
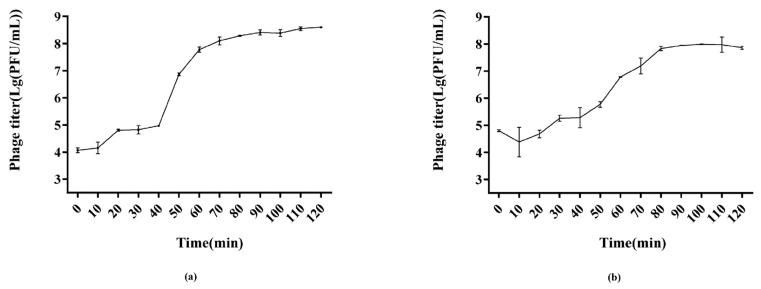
One step growth curves of phages. (**a**) phage PSE-D1. (**b**) phage PST-H1.

**Figure 4 viruses-14-02647-f004:**
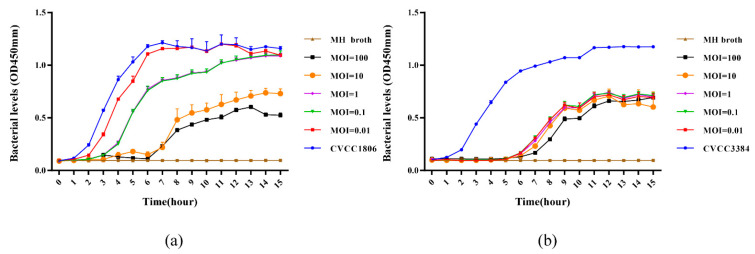
(**a**) Lytic capacity of phage PSE-D1 on *S.* Enteritidis CVCC1806 in LB broth medium. (**b**) Inhibitory effect of phage PST-H1 on *S.* Typhimurium CVCC3384 in LB broth medium. Phage and host strain of each group were added at the same time.

**Figure 5 viruses-14-02647-f005:**
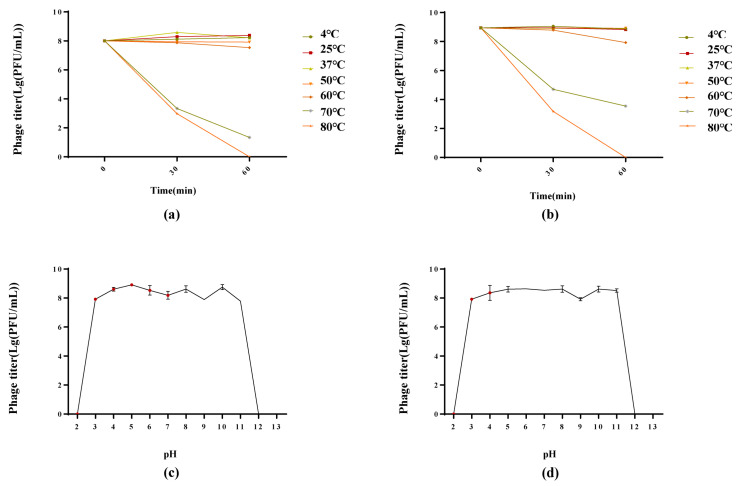
Biological characterization of phages. (**a**) Stability at different pH of PSE-D1. (**b**) Stability at different pH of PST-H1. (**c**) Stability at different temperatures of PSE-D1. (**b**) Stability at different temperatures of PST-H1.

**Figure 6 viruses-14-02647-f006:**
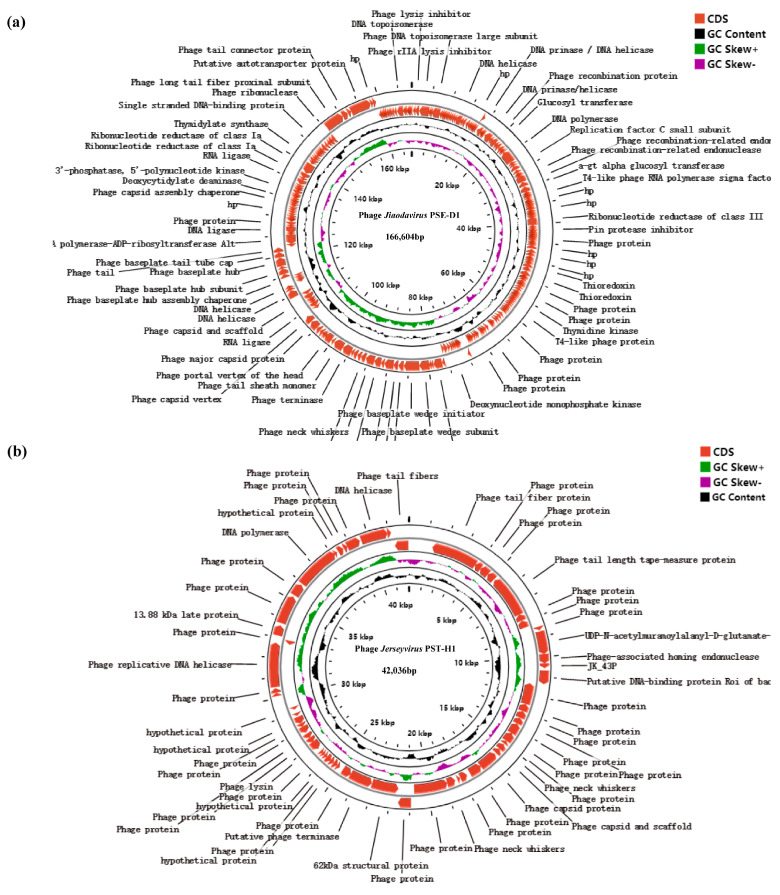
Comparative circular genome maps of phages. (**a**) phage PSE-D1. (**b**) phage PST-H1. Black represents GC content; green and purple represent positive and negative GC skew.

**Figure 7 viruses-14-02647-f007:**
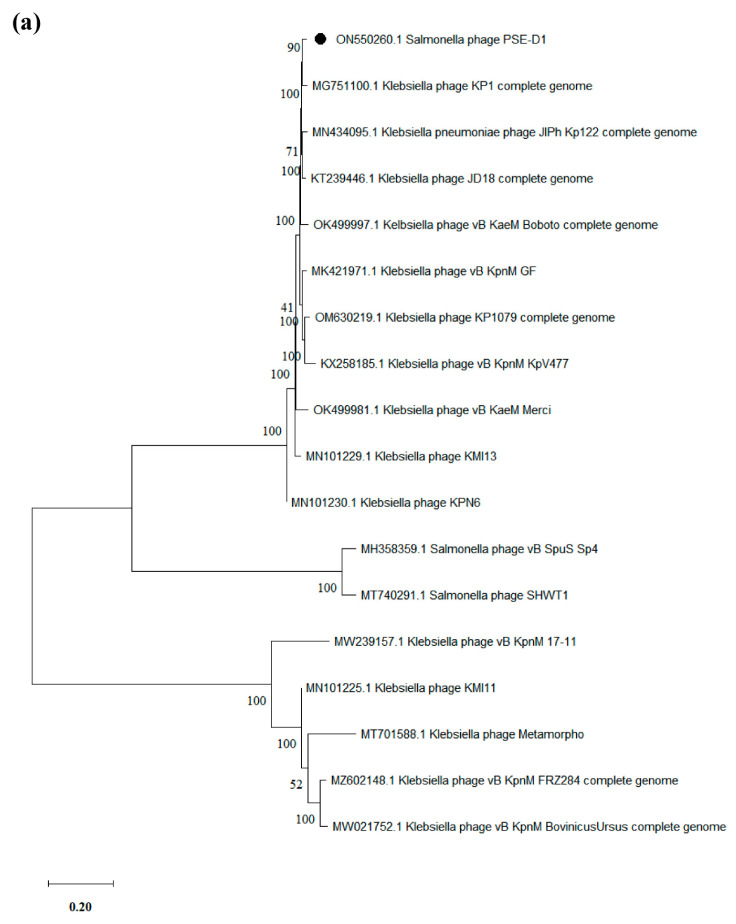
Phylogenetic trees of phages. (**a**) phage PSE-D1. (**b**) phage PST-H1.

**Figure 8 viruses-14-02647-f008:**
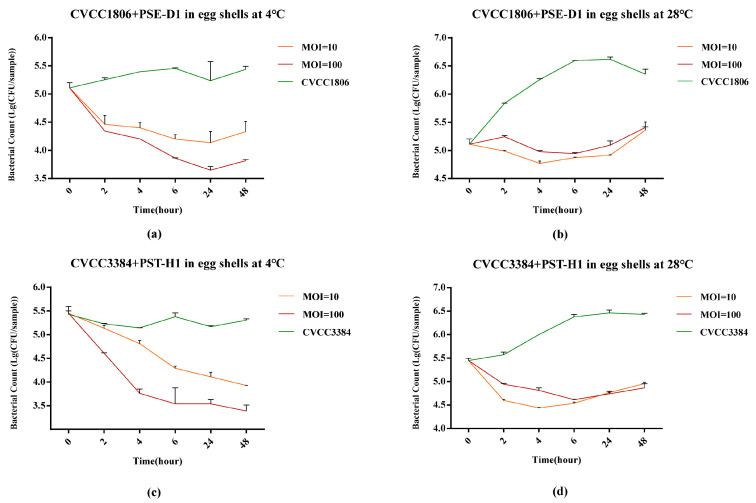
Determination of the antibacterial effect of phages in eggshells at different temperatures. (**a**) CVCC1806+PSE-D1 in eggshells at 4 °C. (**b**) CVCC1806+PSE-D1 in eggshells at 28 °C. (**c**) CVCC3384+PST-H1 in eggshells at 4 °C. (**d**) CVCC3384+PST-H1 in eggshells at 28 °C.

**Figure 9 viruses-14-02647-f009:**
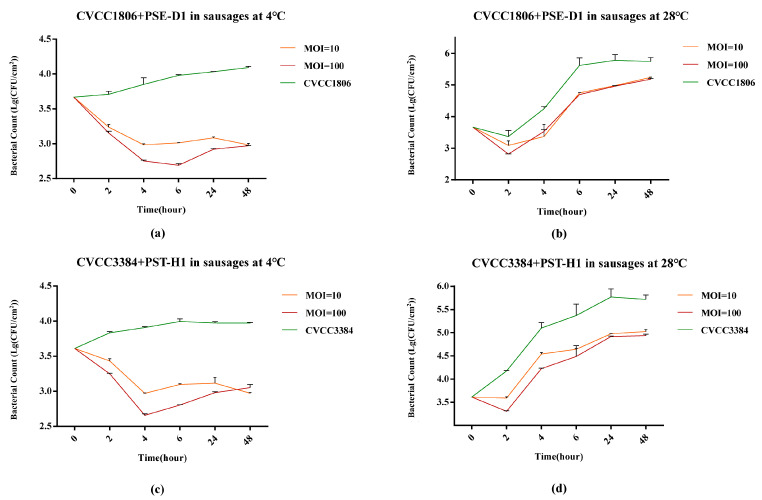
Determination of the antibacterial effect of phages in sausages at different temperatures. (**a**) CVCC1806+PSE-D1 in sausages at 4 °C. (**b**) CVCC1806+PSE-D1 in sausages at 28 °C. (**c**) CVCC3384+PST-H1 in sausages at 4 °C. (**d**) CVCC3384+PST-H1 in sausages at 28 °C.

**Figure 10 viruses-14-02647-f010:**
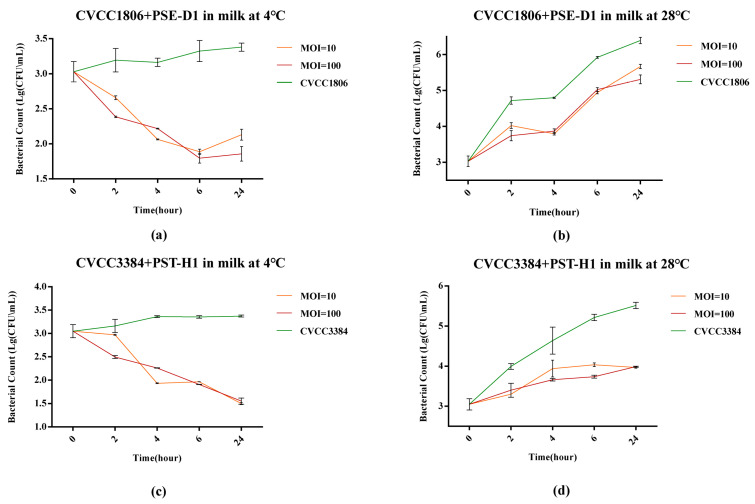
Determination of the antibacterial effect of phages in milk at different temperatures. (**a**) CVCC1806+PSE-D1 in milk at 4 °C. (**b**) CVCC1806+PSE-D1 in milk at 28 °C. (**c**) CVCC3384+PST-H1 in milk at 4 °C. (**d**) CVCC3384+PST-H1 in milk at 28 °C.

**Table 1 viruses-14-02647-t001:** Host strains information of phage PST-H1 and PSE-D1.

Strain Type	Strain Name	Place	Source	Sampling Method	Serotype	PST-H1 Lysis	PSE-D1 Lysis
*Salmonella*	CVCC3384	CVCC	Pig	CVCC	*S.* Typhimurium	+	+
*Salmonella*	CVCC1806	CVCC	Avian	CVCC	*S.* Enteritidis	+	+
*Salmonella*	SX-1014	CVCC	Avian	CVCC	*S.* Pullorum	−	−
*Salmonella*	SF-0923	CVCC	Avian	CVCC	*S.* Pullorum	+	+
*Salmonella*	GXSE-S7	Guangxi	Pig	Intestine	O2, -: Ha: -	−	−
*Salmonella*	GXSE-S4	Guangxi	Pig	Intestine	O7, -: Hc: -	−	−
*Escherichia coli*	GDEC-8	Guangdong	Pig	Anal swab	Undetected	−	+
*Escherichia coli*	GDEC-9	Guangdong	Pig	Anal swab	Undetected	−	+
*Escherichia coli*	GDEC-10	Guangdong	Pig	Anal swab	Undetected	−	−
*Escherichia coli*	GDEC-11	Guangdong	Pig	Anal swab	Undetected	−	−
*Escherichia coli*	GDEC-12	Guangdong	Pig	Anal swab	Undetected	−	−
*Escherichia coli*	GDEC-13	Guangdong	Pig	Anal swab	Undetected	−	−
*Escherichia coli*	GDEC-14	Guangdong	Pig	Anal swab	Undetected	−	+
*Escherichia coli*	GDEC-19	Guangdong	Pig	Anal swab	Undetected	−	+
*Escherichia coli*	GDEC-20	Guangdong	Pig	Anal swab	O127a: K63 (B8)	−	−
*Escherichia coli*	GDEC-27	Guangdong	Pig	Anal swab	Undetected	−	+
*Escherichia coli*	GDEC-28	Guangdong	Pig	Anal swab	O127a: K63 (B8)	−	+
*Escherichia coli*	GDEC-33	Guangdong	Pig	Anal swab	Undetected	+	−
*Escherichia coli*	GDEC-34	Guangdong	Pig	Anal swab	Undetected	+	−
*Escherichia coli*	GDEC-35	Guangdong	Pig	Anal swab	Undetected	−	−
*Escherichia coli*	GDEC-45	Guangdong	Pig	Anal swab	Undetected	+	−
*Escherichia coli*	GDEC-46	Guangdong	Pig	Anal swab	Undetected	+	−
*Escherichia coli*	SCEC-6	Sichuan	Avian	Intestine	Undetected	−	−
*Escherichia coli*	SCEC-7	Sichuan	Avian	Intestine	Undetected	+	−
*Escherichia coli*	SCEC-12	Sichuan	Avian	Intestine	Undetected	−	−
*Escherichia coli*	SCEC-13	Sichuan	Avian	Intestine	Undetected	−	−
*Escherichia coli*	SCEC-14	Sichuan	Avian	Intestine	Undetected	−	−
*Escherichia coli*	SCEC-15	Sichuan	Avian	Intestine	Undetected	−	−
*Escherichia coli*	SCEC-16	Sichuan	Avian	Intestine	Undetected	+	−
*Escherichia coli*	GXEC-1	Guangxi	Avian	Intestine	Undetected	−	−
*Escherichia coli*	GXEC-2	Guangxi	Avian	Intestine	Undetected	−	−
*Escherichia coli*	GXEC-3	Guangxi	Avian	Intestine	Undetected	−	+
*Escherichia coli*	GXEC-4	Guangxi	Avian	Intestine	Undetected	−	−
*Escherichia coli*	GXEC-5	Guangxi	Avian	Intestine	Undetected	−	+
*Escherichia coli*	GXEC-6	Guangxi	Avian	Intestine	Undetected	−	−
*Escherichia coli*	GXEC-7	Guangxi	Avian	Intestine	Undetected	−	+
*Escherichia coli*	GXEC-64	Guangxi	Human	Anal swab	Undetected	−	+
*Escherichia coli*	GXEC-65	Guangxi	Human	Anal swab	O86: K61 (B7)	−	−
*Escherichia coli*	GXEC-66	Guangxi	Human	Anal swab	O86: K61 (B7)	−	+
*Escherichia coli*	GXEC-74	Guangxi	Human	Anal swab	Undetected	−	+
*Escherichia coli*	GXEC-75	Guangxi	Human	Anal swab	Undetected	−	+
*Escherichia coli*	GXEC-80	Guangxi	Human	Anal swab	Undetected	−	−
*Escherichia coli*	GXEC-81	Guangxi	Human	Anal swab	Undetected	+	−
*Escherichia coli*	FJEC-1	Fujian	Human	Anal swab	Undetected	−	−
*Escherichia coli*	FJEC-2	Fujian	Human	Anal swab	Undetected	−	−
*Escherichia coli*	FJEC-3	Fujian	Human	Anal swab	Undetected	−	−
*Escherichia coli*	FJEC-4	Fujian	Human	Anal swab	Undetected	−	−
*Escherichia coli*	FJEC-5	Fujian	Human	Anal swab	Undetected	+	−
*Escherichia coli*	FJEC-6	Fujian	Human	Anal swab	Undetected	−	−
*Escherichia coli*	FJEC-7	Fujian	Human	Anal swab	Undetected	+	−
*Escherichia coli*	FJEC-8	Fujian	Human	Anal swab	Undetected	−	−
*Escherichia coli*	FJEC-9	Fujian	Human	Anal swab	Undetected	−	−
*Escherichia coli*	FJEC-10	Fujian	Human	Anal swab	Undetected	−	−
*Escherichia coli*	FJEC-11	Fujian	Human	Anal swab	Undetected	+	−
*Klebsiella pneumoniae*	GXKP-6	Guangxi	Human	Anal swab	Undetected	−	+
*Klebsiella pneumoniae*	GXKP-13	Guangxi	Human	Anal swab	Undetected	+	+
*Klebsiella pneumoniae*	GXKP-20	Guangxi	Human	Anal swab	Undetected	+	−
*Klebsiella pneumoniae*	GXKP-21	Guangxi	Human	Anal swab	Undetected	−	+
*Klebsiella pneumoniae*	GXKP-49	Guangxi	Human	Anal swab	Undetected	+	+
*Klebsiella pneumoniae*	GXKP-RS3	Guangxi	Human	Anal swab	Undetected	−	+
*Klebsiella pneumoniae*	GXKP-L3	Guangxi	Human	Anal swab	Undetected	−	−
*Klebsiella pneumoniae*	GXKP-L11	Guangxi	Human	Anal swab	Undetected	+	−
*Klebsiella pneumoniae*	GXKP-L15	Guangxi	Human	Anal swab	Undetected	+	−
*Klebsiella pneumoniae*	GXKP-L28	Guangxi	Human	Anal swab	Undetected	+	−
*Klebsiella pneumoniae*	GXKP-L30	Guangxi	Human	Anal swab	Undetected	+	+
*Klebsiella pneumoniae*	GXKP-L34	Guangxi	Human	Anal swab	Undetected	+	−
*Klebsiella pneumoniae*	GXKP-L40	Guangxi	Human	Anal swab	Undetected	+	+
*Klebsiella pneumoniae*	GXKP-L44	Guangxi	Human	Anal swab	Undetected	+	−
*Klebsiella pneumoniae*	GXKP-L45	Guangxi	Human	Anal swab	Undetected	+	+
*Pseudomonas aeruginosa*	SZPA-1	Guangdong	Dog	Skin	Undetected	−	−

“+” indicated that phage could lysis bacteria, “−” indicated no lysis.

## Data Availability

The genome sequence of *Salmonella* phage PSE-D1 and *Salmonella* phage PST-H1 has been submitted to the NCBI Gene Bank and can be accessed by accession numbers ON550260 and ON454038, respectively. The data presented in this study are available on request from the corresponding author.

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
