# Peer review of "Broad-Spectrum Salmonella Phages PSE-D1 and PST-H1 Controls Salmonella in Foods"

_viruses, 2022, doi:10.3390/v14122647_

Round 1

Reviewer 1 Report

This is a well develop study, it is presented properly and is easy to follow. I think the main contribution is the broad range of these phages and the potential they might have. My main concern is about the genome analyses of phages since it seems that it was not achieved the analises of genes associated to lisogenic cycle and for phage therapy is an important requirement. I strongly recomend to develop this analyses for the next version of the article previous to publish.

The others comments are minnor 

1. line 38. it can be confused, and can be seem like food to animals

2. line 83. Please verify the correct taxonomy for Salmonella serotypes throug the article 

3. line 193. Table 1 in the  tittle I suggest  host range results for phages, in the table state the serotype, and also specify what kinf of source was used tissue?, swaps, etc..

4. line 216. change strain by strains

5. line 219 delete obviously

6. line 225. it is not possible to discriminate each line in the figure . state the complete taxonomic name of Salmonella used and in the legend add the time of addition of the phage

7. line 261 CVCC18 i suggest to identify the name of the strain.

Author Response

Thank you very much for your comments and suggestions.

Please see the attachment. All the pages and lines in Responses were marked based on  the clean version of “Track Changes” function of Word in revised maniscript.

Reviewer 2 Report

The results in the article can be improved. In general, the article is interesting. In particular:

3.3.3. Thermal and pH stability of phages.

The stability of the phage can be measured for an extended period, not only a few hours. please improve the results. See for example some articles on Salmonella rissen and phage.

Figure 6. Please enlarge the image

3.5. Application of phages in different foods

The antibacterial activity of phages in different food can be measured for an extended period, especially for sausage and milk. 

Author Response

(The authors gave the same response as above.)

Reviewer 3 Report

Cao et al. isolated two novel broad host range Salmonella phages, characterized them and showed that they have promise to be used in potential applications in food. The authors succeed in their task, the experimental setups are fine and the experiments are well executed. While overall the English is fine, there are a few formulations and repetitions that could be ironed out with the help of a native speaker.

Specific comments:

The authors do not use the most recent nomenclature for Salmonella strains. This nomenclature has changed several times over the years and can be a bit confusing, but the authors should adopt the latest standard. For example:

Salmonella enterica subsp. enterica serovar Typhimurium (the underlined words are italicized)

Is shortened to Salmonella Typhimurium or S. Typhimurium, i.e. the serovar is not italicized and capitalized. The authors inconsistently used S. Typhimurium, S. typhimurium, S. Enteritidis and S. enteritidis and should correct the entire manuscript accordingly.

 Electron Microscopy: in Figure 2 and in the text. The authors mixed the phage family and the tail contractility. Figure 1a shows a Tevenvirinae phage with an elongated icosahedral-shaped head and a “contractile” tail, while Figure 2b shows a Guernseyvirinae phage with an icosahedral head and a long, “non-contractile” tail.

 Discussion, lines 326-328: The burst size of a phage is an important feature, but is not linked to the “lysis capacity”.

Author Response

(The authors gave the same response as above.)
